# Evolution towards simplicity in bacterial small heat shock protein system

Piotr Karaś, Klaudia Kochanowicz, Marcin Pitek, Przemyslaw Domanski, Igor Obuchowski, Barlomiej Tomiczek*, Krzysztof Liberek*

Intercollegiate Faculty of Biotechnology UG-MUG, University of Gdansk, Gdańsk, Poland

**Abstract** Evolution can tinker with multi-protein machines and replace them with simpler single-protein systems performing equivalent functions in an equally efficient manner. It is unclear how, on a molecular level, such simplification can arise. With ancestral reconstruction and biochemical analysis, we have traced the evolution of bacterial small heat shock proteins (sHsp), which help to refold proteins from aggregates using either two proteins with different functions (IbpA and IbpB) or a secondarily single sHsp that performs both functions in an equally efficient way. Secondarily single sHsp evolved from IbpA, an ancestor specialized in strong substrate binding. Evolution of an intermolecular binding site drove the alteration of substrate binding properties, as well as the formation of higher-order oligomers. Upon two mutations in the α-crystallin domain, secondarily single sHsp interacts with aggregated substrates less tightly. Paradoxically, less efficient binding positively influences the ability of sHsp to stimulate substrate refolding, since the dissociation of sHps from aggregates is required to initiate Hsp70-Hsp100-dependent substrate refolding. After the loss of a partner, IbpA took over its role in facilitating the sHsp dissociation from an aggregate by weakening the interaction with the substrate, which became beneficial for the refolding process. We show that the same two amino acids introduced in modern-day systems define whether the IbpA acts as a single sHsp or obligatorily cooperates with an IbpB partner. Our discoveries illuminate how one sequence has evolved to encode functions previously performed by two distinct proteins.

**\*For correspondence:**
bartlomiej.tomiczek@biotech.ug.edu.pl (BT);
krzysztof.liberek@ug.edu.pl (KL)

**Competing interest:** The authors declare that no competing interests exist.

## eLife assessment

This **valuable** study advances our understanding of the evolution of protein complexes and their functions. Through **convincing** experimental and computational methodologies, the authors show that the specialization of protein function following gene duplication can be reversible. The work will be of interest to investigators working in biochemical evolution and those working on heat shock proteins.

## Introduction

Gene birth and loss is a hallmark of protein family evolution, however, molecular determinants and genetic mechanisms enabling that process are not well understood (*Fernández and Gabaldón, 2020*; *Worth et al., 2009*). Gene loss and differential retention of paralogues reshapes the divergence of organisms both in *Animalia*, *Archaea*, and *Bacteria* (*Fernández and Gabaldón, 2020*; *Iranzo et al., 2019*; *Puigbò et al., 2014*). Examples of gene loss often involve adaptive changes in response to changing environmental niches, like loss of genes encoding olfactory receptors in primates (*Demuth and Hahn, 2009*) or differential retention of paralogous genes encoding venom toxins in different rattlesnake lineages (*Dowell et al., 2016*). In prokaryote genomes, gene loss is one of the main evolutionary processes accelerating sequence divergence leading to functional innovations (*Puigbò et al.,*

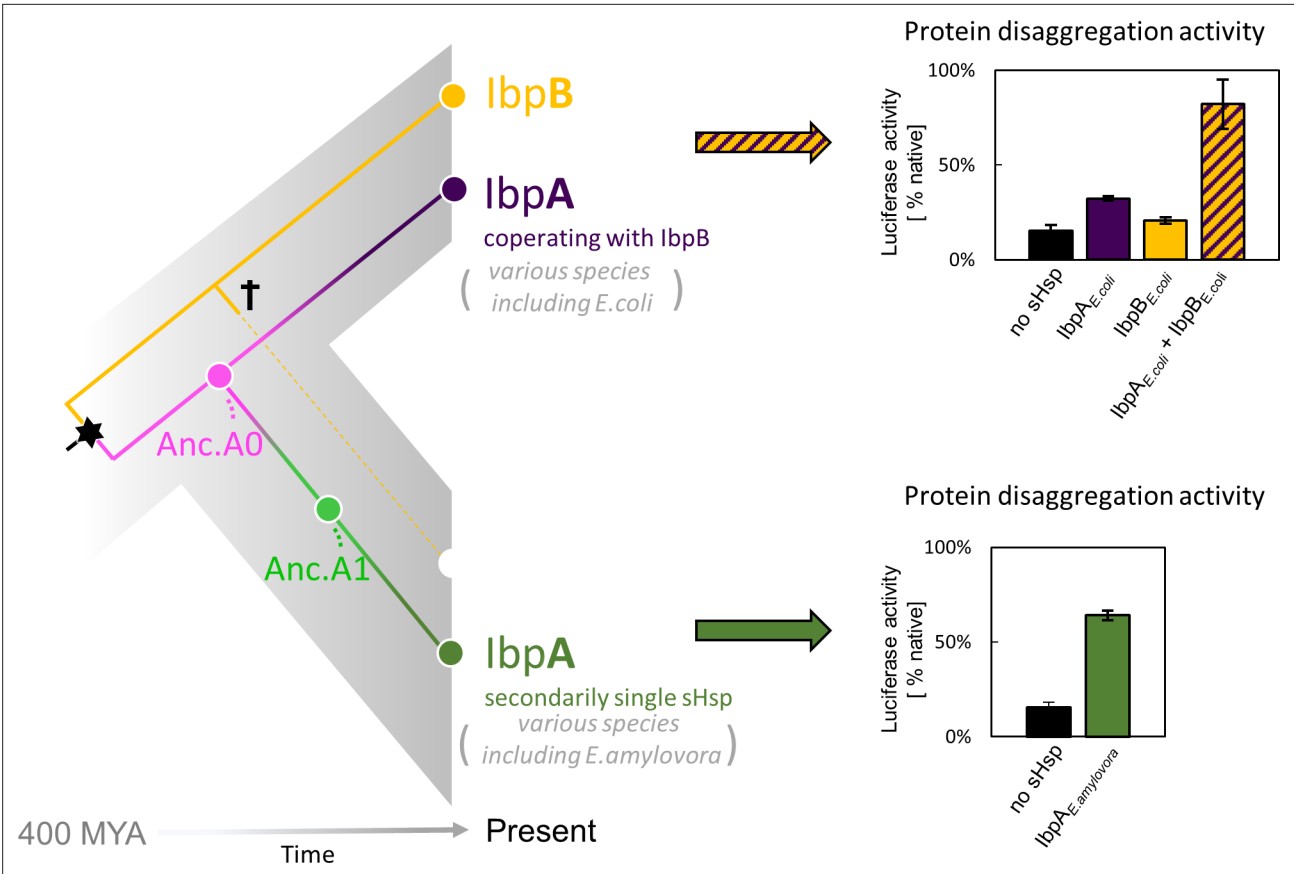

**Figure 1.** small heat shock protein (sHsp) systems in *Enterobacteriacrae* and *Erwiniaceae*. Left - schematic phylogeny of sHsps in *Enterobacterales*. Gene duplication resulting in IbpA + IbpB two-protein system is marked with a star, while the loss of *ibpB* gene in *Erwiniaceae* clade is marked with a cross. AncA$_0$ - reconstructed last common ancestor of IbpA from *Erwiniaceae* and *Enterobacteriaceae*, expressed as a part of two-protein system. AncA$_1$ - reconstructed last common ancestor of secondarily single IbpA from *Erwiniaceae*. Right - representative extant sHsps' ability to stimulate luciferase refolding. sHsps were present during the luciferase thermal denaturation step. Refolding of denatured luciferase was performed by the Hsp70-Hsp100 chaperone system. Activity of luciferase was measured after 1 hr refolding at 25 °C and shown as an average of 3 repeats. Error bars represent standard deviation.

The online version of this article includes the following source data for figure 1:

**Source data 1.** Spreadsheet containing raw data for the luciferase refolding graphs shown in *Figure 1*.

---

*2014*). In complex protein systems, execution of a cellular function can be shared between several proteins. In bacteria, the cost of maintaining additional gene copy is very high and maintaining a low gene count is important for keeping the replication energy costs low (*Kempes et al., 2017*; *Lever et al., 2015*; *Lynch and Marinov, 2015*). Still, it remains unclear how a multi-protein system can undergo simplification. Here, we asked what are the molecular events that enabled the gene loss, and how one of the biochemical functions has been taken over by the other protein. We investigated these questions using sHsp system, which underwent simplification within *Enterobacterales* (which include common bacteria species like *Escherichia coli, Salmonella enterica* and *Erwinia amylovora*) as a model (*Figure 1*).

sHsps are a family of ATP-independent molecular chaperones present in all living organisms with various copy numbers (ten representatives in human) (*Haslbeck and Vierling, 2015*). They bind misfolded proteins and sequester them into refolding – prone assemblies, preventing uncontrolled aggregation and helping to maintain proteostasis at stress conditions. sHsp is composed of a highly conserved α–crystallin domain (ACD), in a form of so-called β–sandwich, flanked by less conserved, unstructured N- and C-terminal regions (*Haslbeck and Vierling, 2015*; *Haslbeck et al., 2019*; *Reinle et al., 2022*). Their smallest functional unit is usually a dimer, formed by the interaction between ACDs of two neighboring sHsps. Stable sHsp dimers in turn tend to form variable and dynamic higher-order

oligomers, stabilized by N- and C-terminal region interactions. Particularly the interaction between the IXI motif, highly conserved in sHsps C-termini, and the cleft formed by β4 and β8 strands of ACD is critical for oligomer formation (*Haslbeck and Vierling, 2015*; *Kennaway et al., 2005*; *Mani et al., 2016*; *Simmons and Ochoterena, 2000*). Oligomers of bacterial sHsps reversibly dissociate into smaller forms when the temperature increases. It is considered their activation mechanism, probably uncovering substrate interaction sites. The mechanism of sHsps' interaction with misfolded substrates is not yet fully understood, but both N-termini and β4–β8 cleft region have been found to play a role in this process (*Basha et al., 2006*; *Fuchs et al., 2009*; *Jaya et al., 2009*; *Lee et al., 1997*; *Reinle et al., 2022*).

In most *Enterobacterales* a two-protein sHsps system exists, consisting of IbpA and IbpB proteins (*Mogk et al., 2003*; *Obuchowski et al., 2019*). IbpA and IbpB have originated via duplication, form a heterodimer partnership, and are functionally divergent from one another (*Obuchowski et al., 2019*; *Piróg et al., 2021*; *Figure 1A and B*). IbpA is specialized in tight substrate binding (sequestrase activity), while IbpB is required for the dissociation of both sHps from the aggregates, a step necessary to initiate Hsp70-Hsp100 dependent substrate disaggregation and refolding (*Obuchowski et al., 2021*; *Ratajczak et al., 2009*). In a subset of *Enterobacterales* (*Erwiniaceae*), as a result of *ibpB* gene loss, the secondarily single-protein sHsp (IbpA) system has emerged (*Figure 1*). The term 'secondarily single' is used in order to distinguish it from single-protein IbpA from clades in which the duplication did not occur (for example *Vibrionaceae*; *Obuchowski et al., 2019*). How did IbpA evolve to become independent of its partner? In this study, using ancestral reconstruction, we identify mutations, which allowed the secondarily single IbpA to be fully functional without its partner in substrate sequestration and handover to Hsp70-Hsp100-mediated disaggregation and refolding.

## Results

### New activity of *Erwiniaceae* IbpA has evolved in parallel to IbpB gene loss

To better understand the evolution of sHsps after gene loss, we reconstructed the IbpA ancestors from before and after the loss of its IbpB partner. This technique uses multiple sequence alignments of modern-day proteins from different species to infer amino acid sequences of its common ancestors (*Ashkenazy et al., 2012*; *Pupko et al., 2002*) and is widely used to investigate various evolutionary questions (*Gaucher et al., 2008*; *Longo et al., 2020*; *Thomson et al., 2005*; *Thornton et al., 2003*). We created a multiple sequence alignment of 77 IbpA sequences from *Enterobacterales* (*Supplementary file 1*), from which we inferred the phylogeny of IbpA using the maximum likelihood method (*Figure 2*, *Supplementary file 2* – phylogenetic tree in Newick format). From that, we inferred ancestral sequences, which have the highest probability of producing the modern-day sequences using the empirical Bayes method (*Ashkenazy et al., 2012*; *Cohen and Pupko, 2011*; *Cohen et al., 2008*; *Pupko et al., 2002*; *Simmons and Ochoterena, 2000*). Next, we resurrected (i.e. expressed and purified) the last ancestor of IbpA present before (AncA$_0$) and after (AncA$_1$) the differential gene loss (*Figure 2*, *Supplementary file 3*).

Similarly, to modern-day IbpA proteins both AncA$_0$ and AncA$_1$ were fully folded, and reversibly deoligomerized into smaller species under elevated temperature (*Figure 3—figure supplement 1*). Moreover, both ancestral proteins were able to sequester aggregating firefly luciferase in sHsp-substrate assemblies. AncA$_0$ exhibited sequestrase activity on the level comparable to IbpA from *Escherichia coli* (IbpA$_{E.\ coli}$). AncA$_1$ was moderately efficient in this process and IbpA from *Erwinia amylovora* (IbpA$_{E.amyl}$) was the least efficient sequestrase (*Figure 3A*). The differences in sequestrase activity were especially pronounced at lower sHsp concentrations. Next, we tested their ability to bind protein aggregates in real time (*Figure 3B*). Ancestral proteins' interaction with the aggregated substrates was stronger than in the case of extant *E. amylovora* IbpA, but weaker than in the case of extant *E. coli* IbpA (*Figure 3B*).

Finally, we asked how the modification of the substrate aggregation process by reconstructed proteins influences subsequent substrate refolding by the Hsp100 and Hsp70 chaperones. AncA$_1$ stimulated luciferase refolding, however, its effectiveness was around half of both analyzed extant sHsp systems (single IbpA from *E. amylovora* or IbpA + IbpB system from *E. coli*), similar to extant



**Figure 2.** IbpA phylogeny in *Enterobacterales*. Phylogeny was reconstructed from 77 IbpA orthologs from *Enterobacterales* using the Maximum Likelihood algorithm with JTT + R3 substitution model. AncA$_0$ - node representing the last common ancestor of IbpA from *Erwiniaceae* and *Enterobacteriaceae*. AncA$_1$ - node representing the last common ancestor of IbpA from *Erwiniaceae*. Bootstrap support is noted for the major nodes. Extant IbpAs from *E. coli* and *E. amylovora* are marked with a red frame. Scale bar - substitutions per position.

IbpA form *E. coli* without its IbpB partner. AncA$_0$, in contrast, inhibited luciferase refolding in comparison to control (no sHsps at substrate aggregation step) (*Figure 3C*).

To test the robustness of our observations, we repeated the analysis for the alternative ancestors, which have the second highest probability to produce the modern-day sequences (AltAll) (*Figure 3—figure supplement 2*, *supplementary file 3* – posterior probabilities) (*Eick et al., 2017*). Both AltAll variants behaved similarly to most likely (ML) variants in reversible deoligomerization, sequestrase activity and stimulation of substrate refolding assays (*Figure 3—figure supplement 3A-D*). However,

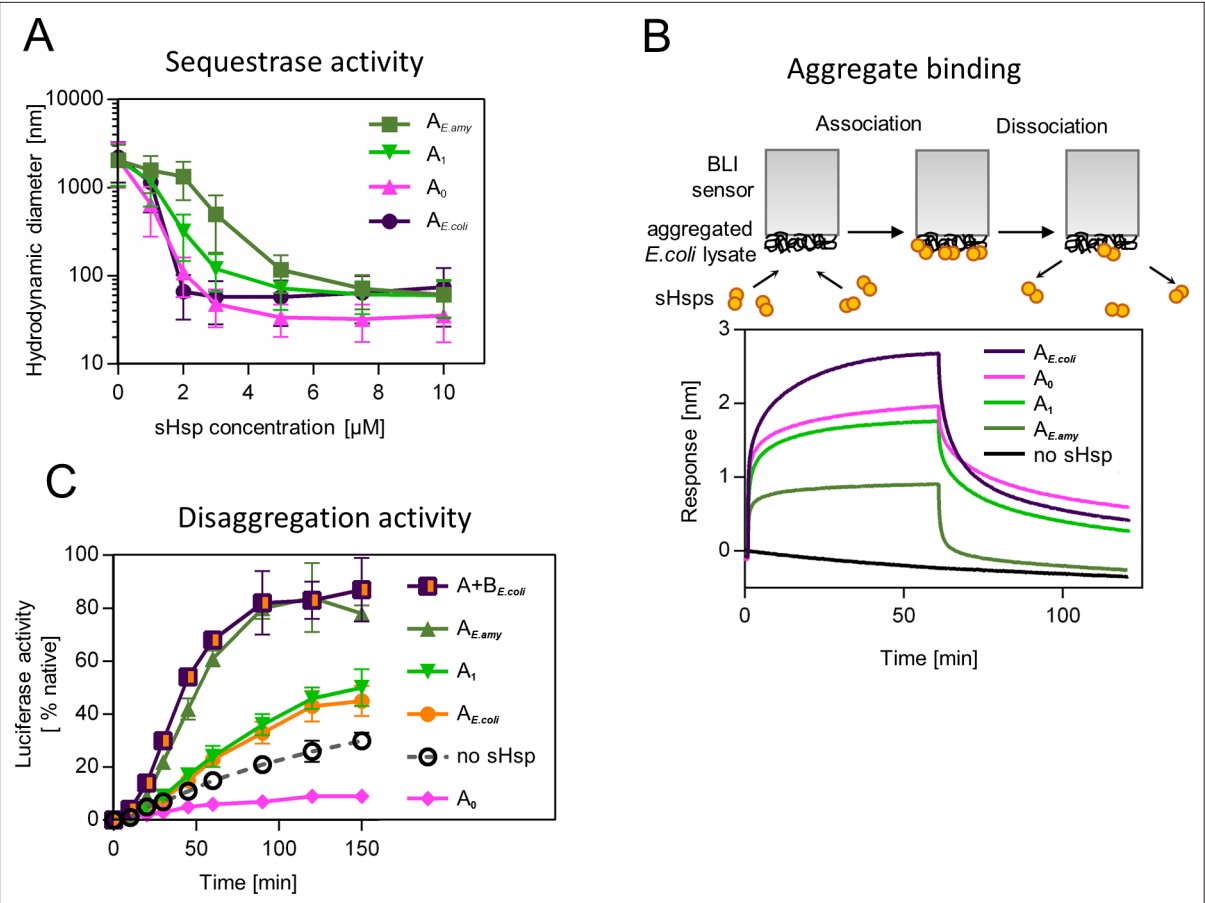

**Figure 3.** Functional changes during the evolution of secondarily single small heat shock protein (sHsp) in *Erwiniaceae*. (**A**) Sequestrase activity of extant and ancestral sHsps. Luciferase was heat denatured in the presence of different concentrations of sHsps and size of formed sHsps-substrate assemblies was measured by DLS. For every measurement, ten subsequent series of ten 10-s runs were averaged and particle size distribution was calculated by fitting to 70 size bins between 0.4 and 10,000 nm. Results are presented as an average hydrodynamic diameter of measured particles weighted by intensity (Z-average), calculated with Malvern Zetasizer Software 7.13. Error bars represent standard deviation obtained from the distribution. (**B**) Binding of extant and ancestral sHsps to heat-aggregated *E. coli* proteins. *E. coli* proteins were heat aggregated and immobilized on a Biolayer interferometry (BLI) sensor. sHsps were heat-activated before the binding step. (**C**) Extant and ancestral sHsps' ability to stimulate luciferase refolding. Experiment was performed at 25 °C. Luciferase activity at each timepoint was shown as an average of 3 repeats. Error bars represent standard deviation.

The online version of this article includes the following source data and figure supplement(s) for figure 3:

**Source data 1.** Spreadsheet containing raw data for the sequestrase activity graph shown in *Figure 3A*.

**Source data 2.** Spreadsheet containing raw data for the binding and dissociation curve shown in *Figure 3B*.

**Source data 3.** Spreadsheet containing raw data for the luciferase refolding graph shown in *Figure 3C*.

**Figure supplement 1.** Both reconstructed proteins reversibly deoligomerize at heat shock temperature, similarly to extant proteins.

**Figure supplement 1—source data 1.** Spreadsheet containing raw data for DLS curves shown in *Figure 3—figure supplement 1*.

**Figure supplement 2.** Amino acid sequences of most likely (ML) and AltAll variants of reconstructed ancestral proteins.

**Figure supplement 3.** AltAll variants of AncA$_0$ and AncA$_1$ have similar properties to most likely (ML) variants.

**Figure supplement 3—source data 1.** Spreadsheet containing raw data for DLS curves shown in *Figure 3—figure supplement 3A*.

**Figure supplement 3—source data 2.** Spreadsheet containing raw data for sequestrase activity measurements shown in *Figure 3—figure supplement 3B*.

**Figure supplement 3—source data 3.** Spreadsheet containing raw data for luciferase refolding graphs shown in *Figure 3—figure supplement 3C*.

**Figure supplement 3—source data 4.** Spreadsheet containing raw data for luciferase refolding graphs shown in *Figure 3—figure supplement 3D*.

the last property (the influence on refolding) required higher Hsp70 system concentration to observe AltAll variants activity (*Figure 3—figure supplement 3C, D*). Together, these data show that reconstructed ML and AltAll ancestors are functional. What is of particular interest, these data clearly point out that the ability of reconstructed sHsps to stimulate Hsp70-Hsp100-dependent substrate refolding arose between $A_0$ and $A_1$ nodes.

We performed a molecular evolution analysis to test for positive selection across IbpA phylogeny using both branch models and branch-site models in codeml (*Jeffares et al., 2015*; *Yang, 1998*; *Yang, 2007*; *Yang and Nielsen, 2002*). The analysis shows a significantly increased ratio of nonsynonymous to synonymous substitutions, after the gene loss, at the branch leading to $A_1$ with both tests. This result indicates that the new IbpA functionality likely arose due to an episode of positive selection rather than genetic drift (*Figure 4A*, *Supplementary file 4 A,B*). The result of the branch-site test indicates possible positive selection acting on all sites substituted at the branch leading to $A_1$ with pp >0.5, therefore, we aimed to identify the minimum number of mutations that are responsible for a change in functional properties of IbpA.

## Identification of residues defining ancestral sHsps activities

In order to identify amino acids responsible for the observed new functionality of $AncA_1$, we compared the sequences of two ancestral proteins, selecting 7 out of 10 substitutions as probable candidates. Three substitutions were removed from the analysis based on the low conservation of these positions in extant proteins (*Figure 4—figure supplement 1*). The remaining seven were introduced into $AncA_0$. Resulting $AncA_0 + 7$ protein stimulated Hsp70-Hsp100-dependent luciferase refolding at the level comparable to $AncA_1$ (*Figure 4B*). To further specify key mutations, we prepared seven additional variants. In each variant, a different position in $AncA_0 + 7$ was reversed to a more ancestral state. The substantial decrease in luciferase refolding stimulation was observed for positions 66 and 109 (*Figure 4B*). Next, each of these substitutions on its own (Q66H or G109D) was separately introduced into $AncA_0$. This was not sufficient to increase $AncA_0$ ability to stimulate luciferase refolding. However, when both substitutions were introduced simultaneously, the resulting sHsp exhibited activity similar to $AncA_1$ (*Figure 4C*). What is more, when in $AncA_1$ these two positions were reversed to $AncA_0$-like state, the resulting sHsp lost the ability to stimulate luciferase refolding (*Figure 4C*). All analyzed proteins, namely $AncA_0 + 7$, $AncA_0$ Q66H G109D and $AncA_1$ H66Q D109G, possess biochemical properties characteristic for sHsps, exhibiting reversible thermal deoligomerization and sequestrase activity (*Figure 4—figure supplement 2* A,B). All these results show that substitutions Q66H and G109D are both sufficient and necessary for the increase in activity observed for ancestral sHsps between $A_0$ and $A_1$ nodes.

## Identified substitutions influence α-crystallin domain properties

Substitutions Q66H and G109D, responsible for gaining single sHsp activity, are located in the ACD within β4 and β8 strands, which form a cleft responsible for the interaction with unstructured C-terminal peptide of the neighboring sHsp dimer (*Figure 5A*). In order to identify possible structural underpinnings of the single sHsp activity we have predicted structures of $AncA_0$ and $AncA_0$ Q66H G109D ACD dimers in complex with C-terminal peptide using AlphaFold2 and in silico mutagenesis and subjected them to 0.5 µs equilibrium molecular dynamics (MD) simulations. Analysis of the C-terminal peptide interface contact probabilities in MD trajectories showed that both substituted residues contact the C-terminal peptide, although the overall contact pattern remain similar upon their introduction (*Figure 5—figure supplement 1*) and no major differences in the overall ACD domains structure were observed (*Figure 5—figure supplement 2*). To explore the possibility that identified substitutions affect the strength of this interaction, we analyzed the binding of purified ACDs of $AncA_0$ and $AncA_0$ Q66H G109D to C-terminal peptide using biolayer interferometry. Titrations of immobilized C-terminal peptides by different ACDs (*Figure 5B*, *Figure 5—figure supplement 3*) allowed us to determine the dissociation constants. These two substitutions increased the $K_{0.5}$ of ACD binding to the C-terminal peptide from 4.3 µM to 7.1 µM at the same time increasing the Hill coefficient of the interaction from 2.3 to 3.7, indicating a modest decrease in affinity, accompanied by an increase in binding cooperativity.

As this interaction is known to play a crucial role in the formation of sHsp oligomers (*Fu et al., 2005*; *Mani et al., 2016*; *Simmons and Ochoterena, 2000*), we used dynamic light scattering to

**Figure 4.** Substitutions at positions 66 and 109 that occurred between nodes $A_0$ and $A_1$ are crucial for ancestral small heat shock proteins (sHsps) to work as a single protein. Luciferase refolding assay was performed as in *Figure 1*. Activity of luciferase was measured after 1 hr refolding at 25 °C and shown as an average of 3 repeats (6 in the case of no sHsp and proteins AncA$_0$ and AncA$_1$ in panel C). Error bars represent standard deviation. (**A**) Schematic phylogeny of *Enterobacterales* IbpA showing increased ratio of nonsynonymous to synonymous substitutions ($\omega$) on the branch between nodes AncA$_0$ and AncA$_1$. Loss of cooperating IbpB is marked on a tree. Value of the Likelihood Ratio Test (LRT) is given for the selection model. (**B**) Identification of substitutions necessary for AncA$_0$ to obtain AncA$_1$-like activity in luciferase disaggregation; seven candidate mutations were introduced into AncA$_0$ (AncA$_0$ + 7); subsequently, in series of six mutants, each of the candidate positions was reversed to a more ancestral state (AncA$_0$ + 6* variants) (**C**) Effect of substitutions at positions 66 and 109 on the ability of AncA$_0$ and AncA$_1$ to stimulate luciferase refolding.

The online version of this article includes the following source data and figure supplement(s) for figure 4:

**Source data 1.** Spreadsheet containing raw data for the luciferase refolding graph shown in *Figure 4B*.

**Source data 2.** Spreadsheet containing raw data for the luciferase refolding graph shown in *Figure 4C*.

**Figure supplement 1.** Amino acid sequence differences between AncA$_0$ and AncA$_1$: Differing positions used in further analysis were marked in red, an differing positions omitted from further analysis due to low conservation in extant *Erwiniaceae* were marked in bold.

**Figure supplement 2.** AncA$_0$ Q66H G109D, AncA$_0$ + 7 and AncA$_1$ H66Q D109G exhibit sequestrase activity and reversibly deoligomerize at heat shock temperature.

**Figure supplement 2—source data 1.** Spreadsheet containing raw data for DLS curves shown in *Figure 4—figure supplement 2A*.

*Figure 4 continued on next page*

*Figure 4 continued*

**Figure supplement 2—source data 2.** Spreadsheet containing raw data for sequestrase activity measurements shown in *Figure 4—figure supplement 2B*.

investigate how Q66H and G109D substitutions influence the size of oligomers formed by AncA$_0$ at different temperatures. In agreement with decreased affinity between ACD and the C-–terminal peptide, we have shown that these substitutions slightly decrease the oligomer size and facilitate AncA$_0$ deoligomerization (*Figure 5—figure supplement 4* A,B).

β4-β8 cleft in certain sHsps, in addition to its interaction with C-terminal peptide, was also shown to participate in interactions with substrates and partner proteins (*Fuchs et al., 2009*; *Jaya et al., 2009*; *Lee et al., 1997*; *Reinle et al., 2022*). Therefore, we tested whether the ACD of AncA$_0$ binds protein aggregates and whether this interaction is influenced by Q66H G109D substitutions. We observed that AncA$_0$ ACD efficiently binds to either aggregated *E. coli* lysate or aggregated luciferase, and this binding was weakened by analyzed substitutions (*Figure 5C*, *Figure 5—figure supplement 5*). This suggests that ACD of bacterial sHsps interacts with the substrate, most likely through the β4-β8 cleft.

These results allow us to conclude that substitutions Q66H and G109D in AncA$_0$ substantially increased the sHsp ability to stimulate Hsp70-Hsp100-dependent substrate refolding by weakening the interaction of β4-β8 cleft with both the C-terminal peptide and the aggregated substrates. Despite its ability to bind aggregated substrates in biolayer interferometry assay, analyzed ACDs do not exhibit sequestrase activity and were unable to positively influence substrate refolding by the Hsp70-Hsp100 system (*Figure 5—figure supplement 5B, C*).

## Identified substitutions define the mode of action of extant sHsps

As more ancestral, AncA$_0$-like state in positions 66 and 109 is conserved in IbpA of *E. coli* while more modern, AncA$_1$-like state is conserved in IbpA of *E. amylovora*, we decided to ask whether this difference is sufficient to explain functional differences between the two extant proteins. Therefore, we introduced AncA$_1$-like substitutions into IbpA$_{E.coli}$ and AncA$_0$-like substitutions into IbpA$_{E.amyl}$. Resulting IbpA$_{E.coli}$ Q66H G109D, in comparison to wild-type IbpA$_{E.coli}$, exhibited an increased ability to stimulate Hsp70-Hsp100-dependent luciferase refolding, as well as a decreased ability to bind aggregated substrates becoming more similar to modern *E. amylovora* IbpA. At the same time, IbpA$_{E.amyl}$ H67Q D110G significantly less efficiently stimulated luciferase refolding, while exhibiting increased ability to bind aggregated substrates in comparison to wild-type IbpA$_{E.amyl}$ (*Figure 6A–C*). Still, both new IbpA variants exhibited properties characteristic of sHsps, namely reversible thermal deoligomerization and sequestrase activity (*Figure 6—figure supplement 1A, B*).

Above results suggest that tight sHsp binding to aggregates negatively affects the subsequent Hsp70-Hsp100-dependent substrate refolding process. It is initiated by binding of the Hsp70 system (DnaK and cochaperones DnaJ and GrpE) to aggregates that require sHsps to be outcompeted from aggregates (*Żwirowski et al., 2017*). To gain insight into the competition between sHsps and Hsp70 we modified the biolayer interferometry experiments and introduced the sensor with sHsps bound to luciferase aggregates into a buffer containing Hsp70 system (*Figure 6D*). Although biolayer interferometry cannot distinguish between proteins bound to the sensor, we took advantage of the differences in the thickness of the protein layers specific for sHsp or Hsp70 binding and also in the binding kinetics. The analysis of the Hsp70 binding to the aggregates covered with sHsps clearly shows that the presence of IbpA$_{E.amyl}$ or IbpA$_{E.coli}$ Q66H G109D on aggregates only weakly inhibits Hsp70 binding (*Figure 6E and H*). In contrast, the inhibition is much more pronounced when IbpA$_{E.amyl}$ H67Q D110G or IbpA$_{E.coli}$ are present on aggregates (*Figure 6F and G*).

All the above experiments indicate that two specific amino acids in positions 66 and 109 in ACD of IbpA proteins define the mode of IbpA activity. Glutamine 66 and glycine 109 are characteristic of IbpA proteins which bind tightly to substrates and thus are not easily outcompeted from the aggregates by Hsp70s. Such IbpAs require IbpB partner cooperation to function properly. Substitutions at these positions to histidine (position 66) and aspartic acid (position 109) allowed for the emergence of a single sHsp which binds to aggregating substrate less tightly and can be outcompeted from the aggregates by Hsp70s in the absence of IbpB.

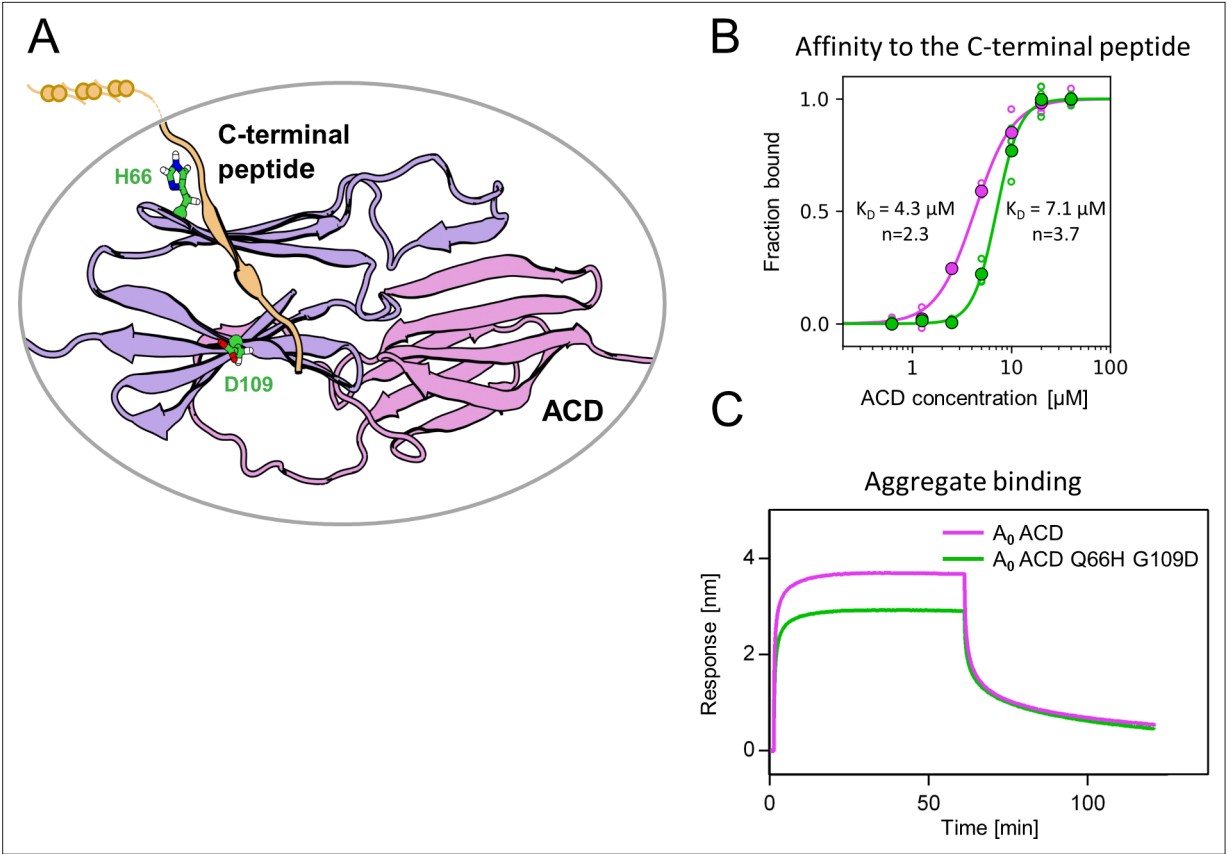

**Figure 5.** Substitutions at positions 66 and 109 decreased the affinity of AncA$_0$ ACD to C-terminal peptide and aggregated substrate. (**A**) Structural model of complex formed by AncA$_0$ Q66H G109D α-crystallin domain dimer (purple and lilac) and AncA$_0$ C-terminal peptide (orange). (**B**) Effect of Q66H G109D substitutions (green) on AncA$_0$ (purple) ACD's affinity to the C-terminal peptide assayed by Biolayer interferometry (BLI). Biolayer thickness at the end of the association step was used to calculate the fraction of bound peptide. Filled circles represent means of triplicate measurements, individual data points are shown as hollow circles and were fitted to the cooperative binding model (Hill equation). Values of fitted binding affinities [K$_{0.5}$] (AncA$_0$ 4.3 ± 0.2 μM, AncA$_0$ Q66H G109D 7.1 ± 0.2 μM) and Hill coefficients [n] (AncA$_0$ 2.3 ± 0.17, AncA$_0$ Q66H G109D 3.7 ± 0.34) are indicated on the plot. (**C**) Effect of Q66H G109D substitutions on AncA$_0$ ACD's affinity to aggregated *E. coli* proteins bound to BLI sensor. Analysis was performed as in *Figure 3A*.

The online version of this article includes the following source data and figure supplement(s) for figure 5:

**Source data 1.** PDB file of the structural model of complex formed by AncA0 Q66H G109D α-crystallin domain dimer and AncA0 C-terminal peptide shown in *Figure 5A* and *Figure 5—figure supplement 2*.

**Source data 2.** Spreadsheet containing raw values of biolayer thickness at the end of the association step (for binding to the C-terminal peptide and nonspecific binding to His$_6$-SUMO), used for fitting the curve shown in *Figure 5B*.

**Source data 3.** Spreadsheets containing raw data for binding and dissociation curves used to obtain biolayer thickness at the end of association curve values shown in *Figure 5—source data 2*.

**Source data 4.** Spreadsheets containing raw data for binding and dissociation curves shown in *Figure 5C*.

**Figure supplement 1.** AncA$_0$ residues 66 and 109 contact the C-terminal peptide, although overall contact pattern remains similar upon introduction of Q66H G109D substitutions.

**Figure supplement 2.** Comparison of the structural models of AncA0 (purple) and AncA0 Q66H G109D (green) α–crystallin domain (ACD) dimers complexed with the C-terminal peptides.

**Figure supplement 2—source data 1.** PDB file of the structural model of complex formed by AncA0 α-crystallin domain dimer and AncA0 C-terminal peptide shown in *Figure 5—figure supplement 2*.

**Figure supplement 3.** Substitutions Q66H G109D decrease affinity of AncA$_0$ ACD to AncA$_0$ C-terminal peptide.

**Figure supplement 4.** Substitutions Q66H G109D influence AncA$_0$ oligomerization.

**Figure supplement 4—source data 1.** Spreadsheet containing raw data for DLS curves shown in *Figure 5—figure supplement 4A–C*.

**Figure supplement 4—source data 2.** Spreadsheet containing raw data for graph shown in *Figure 5—figure supplement 4D*.

*Figure 5 continued on next page*

*Figure 5 continued*

**Figure supplement 5.** AncA$_0$ ACD and AncA$_0$ ACD Q66H G109D can bind to aggregated luciferase, but do not exhibit sequestrase activity or ability to stimulate luciferase refolding.

**Figure supplement 5—source data 1.** Spreadsheet containing raw data for binding and dissociation curves shown in *Figure 5—figure supplement 5A*.

**Figure supplement 5—source data 2.** Spreadsheet containing raw data for sequestrase measurements shown in *Figure 5—figure supplement 5B*.

**Figure supplement 5—source data 3.** Spreadsheet containing raw data for luciferase refolding graphs shown in *Figure 5—figure supplement 5C*.

## Discussion

In this study, we traced, at the molecular level, how the two-protein sHsp system, a part of the cellular protein refolding machinery, underwent simplification in a way that its biochemical functions are performed by a single protein. In most *Enterobacterales* two sHsps (IbpA and IbpB) drive the sequestration of misfolded proteins into the reactivation-prone assemblies (*Obuchowski et al., 2019*). Together, IbpA and IbpB form a functional heterodimer, in which IbpA specializes in substrate binding, preventing the substrates from creating large aggregates (sequestrase activity), while IbpB promotes sHsps dissociation from the aggregates required for subsequent Hsp70-Hsp100-dependent substrate refolding (*Obuchowski et al., 2019*; *Piróg et al., 2021*). We showed that, in parallel to the *ibpB* gene loss in *Erwiniaceae*, new functions of IbpA have emerged, i.e., a lower substrate sequestrase activity, which correlates with the efficient substrate refolding. We have identified two amino acid substitutions (Q66H and G109D) responsible for this new IbpA functionality. Selection analysis shows that these two substitutions were likely driven by positive selective pressure. This indicates that this change possibly had an adaptive character in the ancestral background. It is important to note, however, that models used for the analysis do not account for the variation of synonymous substitution rate and multinucleotide substitution events, which in some cases might lead to false positive results (*Lucaci et al., 2023*). Because of that, the alternative hypothesis that substitutions Q66H and G109D occurred in the common ancestor of *Erwiniaceae* due to genetic drift, enabling the subsequent loss of IbpB gene, cannot be fully discounted. Functional differences observed between modern-day sHsps from *E. coli* and *E. amylovora* are at least partially defined by the presence of specific amino acids in these two positions and can be diminished by their swapping between extant proteins. Their occurrence in the last common ancestor of *Erwiniaceae* IbpA resulted in a decreased affinity of ACD's β4-β8 cleft to aggregated substrates as well as to the C-termini of the other sHsps. These interactions might be of particular importance for the stabilization of sHsps on a surface of sequestered aggregated substrates leading to the formation of so-called protective shell preventing further uncontrolled aggregation. Apparent role of C-termini and ACD interaction is in agreement with earlier studies, showing that the addition of free C-terminal peptide causes *E. coli* IbpA and IbpB dissociation from the outer shell of sHsp - substrate complex (*Żwirowski et al., 2017*). It was also shown that, in case of Hsp 16.6 from cyanobacterium *Synechocystis*, substitutions that slightly weakened the interaction between ACD and C-termini lead to increased stimulation of luciferase refolding. However, abolishing this interaction resulted in a non-functional protein, most likely due to the loss of the sequestrase activity (*Giese and Vierling, 2002*). Destabilization of the protective shell of secondarily single IbpA by weakening these interactions had an effect functionally analogous to a role of IbpB in the two-protein system, facilitating IbpA dissociation from the substrate (*Obuchowski et al., 2019*). Two identified substitutions also weaken the ACD interaction with aggregated substrates which is an additional factor shifting the sHsps balance towards dissociation, a step necessary to initiate Hsp70-Hsp100 dependent disaggregation and refolding.

Our results show how ACD substitutions can fine-tune sHsp system by exerting pleiotropic effects on ACD-C-terminal peptide and ACD-substrate interactions. These relatively small changes strongly influence the effectiveness of sHsp functioning in complex process of aggregated protein rescue by molecular chaperones. It might be particularly important in the case of a conserved interaction, like the one between C-terminal peptide and ACD, when excessive changes of affinity may be detrimental to the overall protein function (*Giese and Vierling, 2002*). Our approach enabled us to find functional residues in sHsp system, which would not have been possible by using conventional mutagenesis and highlights the importance of using the vertical approach in biochemical studies. This study closely follows the evolutionary process, in which mutations in one of the two cooperating proteins tinker it

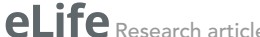

**Figure 6.** Differences at positions 66 and 109 determine functional differences between extant IbpA proteins from *E. coli* and *E. amylovora*. (**A**) Effect of substitutions at position 66 and 109 (and homologous) on the ability of IbpA from *E. amylovora* and *E. coli* to stimulate luciferase refolding. Assay was performed as in ***Figure 1B***. Activity of luciferase was measured after 1 h refolding at 25 °C and shown as an average of 3 repeats. Error bars represent standard deviation. (**B, C**) Effect of substitutions at analyzed positions on binding of IbpA from *E. coli* (**B**) and *E. amylovora* (**C**) to heat - aggregated *E.*

*Figure 6 continued on next page*

*Figure 6 continued*

*coli* proteins. Assay was performed as in 3 A. (**D–H**) Effect of substitutions at analyzed positions on inhibition of Hsp70 system binding to aggregates by extant small heat shock proteins (sHsps) (**D**) Experimental scheme. (**E–H**) Aggregate-bound sHsps differently inhibit Hsp70 binding. Biolayer interferometry (BLI) sensor with immobilized aggregated luciferase and aggregate bound sHsps was incubated with Hsp70 or buffer (spontaneous dissociation curve).Dark gray traces (dashed line) represent Hsp70 binding to immobilized aggregates in the absence of sHsps. Results are presented as an average of 5 (in the case of IbpA $_{E.\ coli}$ in panel G), 4 (in the case of IbpA $_{E.\ coli}$ Q66H G109d with Hsp70 in panel H), 2 (in the case of IbpA $_{E.\ coli}$ Q66H G109d without Hsp70 in panel H) or 3 repeats (in the case of the remaining curves). Error bars (shades) represent standard deviation.

The online version of this article includes the following source data and figure supplement(s) for figure 6:

**Source data 1.** Spreadsheet containing raw data for the luciferase refolding graph shown in *Figure 6A*.

**Source data 2.** Spreadsheet containing raw data for binding and dissociation curve shown in *Figure 6B*.

**Source data 3.** Spreadsheet containing raw data for binding and dissociation curve shown in *Figure 6C*.

**Source data 4.** Spreadsheet containing raw data for binding and dissociation curves shown in *Figure 6E*.

**Source data 5.** Spreadsheet containing raw data for binding and dissociation curves shown in *Figure 6F*.

**Source data 6.** Spreadsheet containing raw data for binding and dissociation curves shown in *Figure 6G*.

**Source data 7.** Spreadsheet containing raw data for binding and dissociation curves shown in *Figure 6H*.

**Figure supplement 1.** IbpA$_{E.coli}$ H66Q G109D and IbpA$_{E.amyl}$ H67Q D110G exhibit sequestrase activity and reversibly deoligomerize at heat shock temperature: (**A**) Reversible deoligomerization of IbpA$_{E.\ coli}$ H66Q G109D and IbpA$_{E.amyl}$ H67Q D110 at heat shock temperature.

**Figure supplement 1—source data 1.** Spreadsheet containing raw data for DLS curves shown in *Figure 6—figure supplement 1A*.

**Figure supplement 1—source data 2.** Spreadsheet containing raw data for sequestrase activity measurements shown in *Figure 6—figure supplement 1B*.

to a point where it becomes independent of its partner, enabling the simplification of a more complex system by partner loss while maintaining its overall function.

Following, with molecular precision, the genetic events associated with the gene loss allowed us to answer several questions about protein family evolution. The first question concerns how the lost function is incorporated into a remaining partner protein. The results of our experiments indicate that even though the primary function of a partner protein is maintained, it is altered so that it does not interfere with the newly incorporated one. In our case substrate binding properties, as well as the formation of higher-order oligomers were altered in a way to keep the sequestrase function maintained but allow for more efficient stimulation of Hsp70-Hsp100-dependent substrate refolding. The second question concerns the context-dependence of mutations in protein evolution. We successfully transplanted the mutations that appeared in *E. amylovora* into *E. coli* IbpA ortholog, artificially creating an efficient single protein system from a protein that normally needs a partner for efficient substrate refolding. In contrast to other studies (*Mogk et al., 2003*) the context of *E. coli* IbpA protein did not influence the ability of IbpA$_{E.coli}$ Q66H G109D to work without IbpB partner. This brings the question about the accessibility of adaptive solutions. In case of *E. coli* IbpA, there was no need for adaptive changes because the gene loss did not occur in this clade, suggesting that the loss of a protein can push another one towards an adaptation, which leads to finding efficient molecular innovations.

# Materials and methods

## Key resources table

| Reagent type (species) or resource | Designation | Source or reference | Identifiers | Additional information |
|---|---|---|---|---|
| Software, algorithm | Clustal Omega | Clustal Omega | RRID:SCR_001591 | European Bioinformatics Institute |
| Software, algorithm | iq-tree | IQ-TREE | | |
| Software, algorithm | FastML | FastML | RRID:SCR_016092 | Tel Aviv University |
| Software, algorithm | codeml | codeml | RRID:SCR_014932 | Part of PAML software package |
| Software, algorithm | Gromacs 2019.2 | GROMACS | RRID:SCR_014565 | |
| Software, algorithm | AlphaFold-Multimer | AlphaFold-Multimer | | ColabFold implementation |

*Continued on next page*

*Continued*

| Reagent type (species) or resource | Designation | Source or reference | Identifiers | Additional information |
|---|---|---|---|---|
| Software, algorithm | PyMol | PyMol | RRID:SCR_000305 | |
| Recombinant DNA reagent | pET3a – $AncA_0$ (plasmid) | This paper | | pET3a plasmid carrying $AncA_0$ gene was ordered from Genscript on the basis of reconstructed amino acid sequence. For the sequence of synthesized gene see *Supplementary file 5* |
| Recombinant DNA reagent | pET3a – $AncA_1$ (plasmid) | This paper | | pET3a plasmid carrying $AncA_1$ gene was ordered from Genscript on the basis of reconstructed amino acid sequence. For the sequence of synthesized gene see *Supplementary file 5* |
| Recombinant DNA reagent | pET3a – $AncA_{0alt\_all}$ (plasmid) | This paper | | pET3a plasmid carrying $AncA_{0alt\_all}$ gene was ordered from Genscript on the basis of reconstructed amino acid sequence For the sequence of synthesized gene see *Supplementary file 5* |
| Recombinant DNA reagent | pET3a – $AncA_{1alt\_all}$ (plasmid) | This paper | | pET3a plasmid carrying $AncA_{1alt\_all}$ gene was ordered from Genscript on the basis of reconstructed amino acid sequence. For the sequence of synthesized gene see *Supplementary file 5* |
| Recombinant DNA reagent | pET3a – $AncA_0 + 7$ (plasmid) | This paper | | pET3a plasmid carrying $AncA_0 + 7$ gene was ordered from Genscript on the basis of reconstructed amino acid sequence. For the sequence of synthesized gene see *Supplementary file 5* |
| Recombinant DNA reagent | pET3a – $IbpA_{Ea}$ (plasmid) | This paper | | pET3a plasmid carrying gene encoding *Erwinia amylovora* IbpA protein was ordered from Genscript on the basis of amino acid sequence obtained from PDB database (accession number E5BAR7). For the sequence of synthesized gene see *Supplementary file 5* |
| Recombinant DNA reagent | pET28a-$AncA_0$ ACD (plasmid) | This paper | | pET28a plasmid carrying $AncA_0$ ACD gene was ordered from Genscript on the basis of reconstructed amino acid sequence. For the sequence of synthesized gene see *Supplementary file 5* |
| Recombinant DNA reagent | pET28a-$AncA_0$ Q66H H109D ACD (plasmid) | This paper | | pET28a plasmid carrying $AncA_0$ ACD Q66H G109D gene was ordered from Genscript on the basis of reconstructed amino acid sequence. For the sequence of synthesized gene see *Supplementary file 5* |
| Recombinant DNA reagent | Champion pET SUMO Expression System (plasmid) | Thermo Fisher Scientific | Cat. # K30001 | |
| Recombinant DNA reagent | pET28a – $His_6$-SUMO-C-peptide (plasmid) | This paper | | pET28a plasmid carrying $His_6$-SUMO-C-peptide gene was ordered from Genscript on the basis of reconstructed amino acid sequence of $AncA_0$ C-terminal peptide. For the sequence of synthesized gene see *Supplementary file 5* |
| Peptide, recombinant protein | OuantiLum Recombinant Luciferase | Promega | Cat. # E1701 | |
| Peptide, recombinant protein | Creatin Kinase | Roche | Cat. # 10127566001 | |
| Peptide, recombinant protein | DnaK | DOI:10.1016 /j. jmb.2008.12.009 | | DnaK protein from *E. coli* |
| Peptide, recombinant protein | DnaJ | DOI:10.1016 /j. jmb.2008.12.009 | | DnaJ protein from *E. coli* |
| Peptide, recombinant protein | ClpB | DOI:10.1016 /j. jmb.2008.12.009 | | ClpB protein from *E. coli* |

*Continued on next page*

*Continued*

| Reagent type (species) or resource | Designation | Source or reference | Identifiers | Additional information |
|---|---|---|---|---|
| Peptide, recombinant protein | IbpA$_{Ec}$ | DOI:10.1016 /j. jmb.2008.12.009 | | IbpA protein from *E. coli* |
| Peptide, recombinant protein | IbpB$_{Ec}$ | DOI: 10.1016 /j. jmb.2021.167054 | | IbpB protein from *E. coli* |
| Peptide, recombinant protein | GrpE | DOI: j.jmb.2008.12.009 | | GrpE protein from *E. coli* |
| Peptide, recombinant protein | His-tagged luciferase | DOI: 10.1371/journal.pgen. 1008479 | | |
| Commercial assay or kit | Luciferase Assay System | Promega | Cat. # E151A, E152A | |
| Strain, strain background (*Escherichia coli*) | *E. coli* BL21 (DE3) | ther | | Laboratory collection |

## Reconstruction of IbpA phylogeny

Amino acid sequences of 77 IbpA orthologs from *Enterobacterales* were obtained from NCBI and UniProt databases and aligned using Clustal Omega (*Sievers et al., 2011*). Alignment was trimmed manually. JTT + R3 was identified as the best-fit model by iq-tree, using Bayesian Information Criterion, and was used in the analysis (*Kalyaanamoorthy et al., 2017*; *Nguyen et al., 2015*). The phylogenetic tree was inferred using iq-tree on the basis of 328 iterations of ML search with 100 rapid bootstraps replicates (*Nguyen et al., 2015*).

## Reconstruction of ancestral IbpA amino acid sequences

Ancestral sequence reconstruction was performed on the basis of multiple sequence alignment of 77 amino acid sequences of IbpA orthologs from *Erwiniaceae* and *Enterobacreriaceae* as well as a phylogenetic tree of those orthologs (see above). Marginal reconstruction of ancestral sequences was performed with FastML program based on ML algorithm and Bayesian approach using JTT substitution matrix with gamma parameter (*Ashkenazy et al., 2012*; *Cohen and Pupko, 2011*; *Cohen et al., 2008*; *Jones et al., 1992*; *Pupko et al., 2002*; *Simmons and Ochoterena, 2000*).

Alternative ancestral sequences for AncA$_0$ and AncA$_1$ proteins were obtained by substituting the most likely amino acid on every uncertain position (defined as a position with more than one amino acid with posterior probability ≥0.2) with the amino acid with the second highest posterior probability (*Eick et al., 2017*).

## Analysis of natural selection

Analysis of natural selection was performed using codeml. First, Pal2Nal was used to obtain codon alignment based on the multiple sequence alignment of amino acid sequences of IbpA orthologs from *Enterobacterales* (see above) as well as corresponding nucleotide sequences obtained from the NCBI database. Resulting codon alignment was then trimmed manually and used together with the phylogenetic tree obtained earlier (see above) for the selection analysis.

For branch model analysis, models M0 (null hypothesis) and two-ratio (with either AncA$_0$-AncA$_1$ branch or *Erwiniaceae* clade as foreground) were used. For branch-site model analysis, models A null (null hypothesis) and A were used, with foreground branches selected as above. Statistical significance of different models was estimated with Likelihood Ratio Test (LRT) (*Jeffares et al., 2015*; *Yang, 1998*; *Yang, 2007*; *Yang and Nielsen, 2002*).

## Protein purification

### Purification of IbpA proteins

pET3a plasmids containing *AncA$_0$*, *AncA$_1$*, *AncA$_0$ + 7*, *AncA$_{0\ alt\_all}$*, *AncA$_{1alt\_all}$*, and *IbpA$_{Ea}$* genes were ordered from GeneScript (*Supplementary file 5*). Point mutations were introduced using site-directed mutagenesis and confirmed by sequencing. Proteins were overproduced in *E. coli* BL21(DE3). Cells were then lysed by sonication in Qsonica sonicator (13% amplitude, 2 min 30 s process time, 15 s

pulse-ON time, 45 s pulse-OFF time) in lysis buffer L1 (50 mM Tris pH 7.5, 50 mM NaCl, 5 mM EDTA, 10% glycerol, 5 mM β-mercaptoethanol). Insoluble fraction containing proteins of interest was separated by centrifugation (75,000 x g, 30 min, 4 °C) and resolubilized in buffer A (40 mM Tris pH 7.5, 50 mM NaCl, 10% glycerol, 5 mM β-mercaptoethanol, 6 M urea) and then centrifuged (75,000 x g, 30 min, 4 °C). Supernatant was loaded on Q-Sepharose chromatography column equilibrated with buffer A and eluted in 50 mM–500 mM NaCl gradient. Fractions containing proteins of interest were then dialyzed to buffer B (40 mM Tris pH 8.5, 50 mM NaCl, 10% glycerol, 5 mM β-mercaptoethanol) and loaded on Q-Sepharose chromatography column equilibrated with buffer B. Flow-through fraction was collected and dialyzed to buffer C (50 mM Tris pH 7.5, 150 mM KCl, 5% (v/v) glycerol, 5 mM β-mercaptoethanol).

### Purification of ACD domains

ACDs of $IbpA_{Ec}$, $AncA_0$, and $AncA_0$ Q66H G109D were purified as described previously (*Piróg et al., 2021*) and as a final step dialyzed to buffer G (50 mM Tris pH 7.5, 150 mM KCl, 5 mM β-mercaptoethanol).

### Purification of $His_6$-SUMO and $His_6$-SUMO-C-terminal peptide of $AncA_0$ construct

$His_6$-SUMO was purified using the Champion pET SUMO Expression System. pET28a plasmid containing gene encoding $His_6$-SUMO fused with C-terminal peptide of $AncA_0$ (PEAMKPPRIEIN) was ordered from GeneScript (*Supplementary file 5*). Proteins were overproduced in *E. coli* BL21(DE3). Cells were then lysed by sonication in Qsonica sonicator (20% amplitude, 2 min process time, 5 s pulse-ON time, 10 s pulse-OFF time) in lysis buffer L2 (40 mM Tris pH 7.5, 100 mM NaCl, 10% glycerol, 10 mM imidazole, 2 mM β-mercaptoethanol). Insoluble fractions were separated by centrifugation for 30 min at 70,000 x g and supernatants, containing proteins of interest, were incubated for 1 hr with Ni-NTA resin equilibrated with buffer L2. Resins were then washed with the buffer D (40 mM Tris pH 7.5, 100 mM NaCl, 10% glycerol, 40 mM imidazole, 2 mM β-mercaptoethanol) Proteins of interest were eluted from the columns with the buffer E (40 mM Tris pH 7.5, 100 mM NaCl, 10% glycerol, 400 mM imidazole, 2 mM β-mercaptoethanol) and then dialyzed to buffer C (as above).

DnaK, DnaJ, GrpE, ClpB, and $IbpA_{Ec}$ proteins were purified as described previously (*Ratajczak et al., 2009*). $IbpB_{Ec}$ protein was purified as described previously (*Piróg et al., 2021*). His-tagged luciferase used for BLI measurements was purified as described previously (*Obuchowski et al., 2019*).

Purity of purified proteins was assessed with SDS-PAGE electrophoresis with Coomassie Blue staining. Protein concentrations were measured using the Bradford reaction, with Bovine Serum Albumin as a standard. In the case of $His_6$-SUMO fused with C-terminal peptide of $AncA_0$, concentration was measured with SDS-PAGE electrophoresis with Coomassie Blue staining coupled with densitometric analysis with Bovine Serum Albumin used as a standard.

OuantiLum Recombinant Luciferase was purchased from Promega. Creatin Kinase from rabbit muscle was purchased from Roche.

### Luciferase refolding assay

1,5 µM recombinant firefly luciferase in buffer F (50 mM Tris pH 7.5, 150 mM KCl, 20 mM $MgCl_2$, 2.5 mM DTT) was denatured by incubation for 10 min at 44 °C alone or in the presence of 10 µM sHsps (3 uM $IbpA_{Ec}$ + 7 µM $IbpB_{Ec}$ in the case of two-protein system from *E. coli*). Denatured luciferase was then incubated at 25 °C with Hsp70 system (1 µM DnaK, 0.4 µM DnaJ, and 0.3 µM GrpE), 2 µM ClpB, and ATP regeneration system (5 mM ATP, 0.1 mg/ml creatine kinase and 18 mM creatine phosphate). For experiment presented in *Figure 3—figure supplement 3D*, higher concentration of the Hsp70 system was used (2 µM DnaK, 0.8 µM DnaJ, and 0.6 µM GrpE). At different timepoints luciferase activity was measured with GLOMAX 20/20 luminometer, using the Luciferase Assay System from Promega. Results are presented as averages of at least three independent repeats ± standard deviation.

### DLS measurements

Dynamic Light Scattering measurements were performed using Malvern Instruments ZetaSizer Nano S instrument, at 40 µl sample volume, scattering angle of 173° and wavelength of 633 nm. For every measurement, a minimum of 10 subsequent series of ten 10 s runs were averaged and particle



size distribution was calculated by fitting to 70 size bins between 0.4 and 10,000 nm, as previously described (*Żwirowski et al., 2017*).

For reversible deoligomerization assay, size of oligomers formed by 10 µM sHsps in buffer F (50 mM Tris pH7.5, 150 mM KCl, 20 mM MgCl$_2$, 2.5 mM DTT) were measured by DLS. First measurement was performed at 25 °C and then the sample was heated to 44 °C, cooled to 25 °C, heated to 44 °C and cooled to 25 °C, with measurements performed after each change in temperature. Results are presented as size distribution by volume.

For measuring the influence of substitutions on oligomer formation, the size of oligomers formed by either 10 µM AncA$_0$ or 10 µM AncA$_0$ Q66H G109D in buffer F (as above) was measured by DLS at 25, 27, 29, 31, 33, 35, 37, 39, 41, 43, and 45°C. Results were presented as size distribution by intensity (for temperatures 25 °C, 35°C, and 45°C) or as an average hydrodynamic diameter corresponding to the maximum of a dominant peak of size distribution by volume plotted against temperature ± standard deviation.

For assembly formation (sequestrase) assay, 1.5 µM firefly luciferase in buffer F was denatured alone or in the presence of different sHsp concentrations by incubation at 44 °C for 10 min. Size of obtained luciferase aggregates was then measured by DLS at 25 °C. Results are presented as an average hydrodynamic diameter of measured particles weighted by intensity (Z-average) ± standard deviation.

## Biolayer interferometry (BLI) measurements

sHsps interactions with aggregated luciferase or aggregated *E. coli* lysate were measured using the Octet K2 system. Anchoring layer of his-tagged luciferase was attached to Octet NTA Biosensors by 5 min incubation in 0.6 mg/ml his-tagged luciferase in denaturing conditions in buffer UF (50 mM Tris pH 7.5, 4.5 M urea, 150 mM KCl, 20 mM MgCl$_2$, 2.5 mM DTT) at 25 °C with 350 rpm shaking. Sensors were then incubated for 5 min as above in buffer H (50 mM Tris pH 7.5, 150 mM KCl, 20 mM MgCl$_2$, 5 mM β-mercaptoethanol) to remove urea and unbound luciferase. The protein aggregate was then formed on the sensor by incubation for 10 min in 0,5 mg/ml His-tagged luciferase or 0.2 mg/ml *E. coli* lysate in buffer H at 44 °C (in case of the luciferase) or 55 °C (in case of the lysate). Sensors were then again incubated in buffer H for 5 min at 25 °C with 350 rpm shaking to remove excess protein. Sensors with attached aggregate were placed in the Octet system in H buffer for 60 s baseline measurement and then placed for 1 hr in 5 µM sHsp solution in H buffer to measure sHsps association. Sensors were then moved for 1 hr to buffer H to measure protein dissociation. Measurements were performed with 1000 rpm shaking at 44 °C (in case of full-length sHsps) or at 25 °C (in case of ACDs). Full-length proteins were preincubated at 44 °C for 10 min before measurement.

sHsps displacement by Hsp70 system was measured using ForteBio BLItz. Sensors with attached aggregates were prepared as described above, with buffer F (50 mM Tris pH 7.5, 150 mM KCl, 20 mM MgCl$_2$, 2.5 mM DTT) instead of buffer H. Baseline biolayer was measured for 60 s. Sensors were then placed in 5 µM sHsp solution in buffer F, previously preincubated for 10 min at 44 °C. sHsps association was measured for 10 min. Sensors were then moved to either buffer F or Hsp70 system in buffer F (0.7 µM DnaK, 0.28 µM DnaJ, 0.21 µM GrpE, 5 mM ATP, 0.1 mg/ml creatine kinase, 18 mM creatine phosphate). Hsp70 system binding and sHsps dissociation were measured for 1 hr. Measurements were performed at room temperature with 2000 rpm shaking.

ACD interactions with C-terminal peptide were measured using Octet K2 system. Octet NTA Biosensors were placed in buffer C (50 mM Tris pH 7.5, 150 mM KCl, 5% glycerol, 5 mM β-mercaptoethanol) and baseline signal was measured for 60 s. Sensors were then placed in 2.5 µM His$_6$-SUMO-C-peptide solution in buffer C and incubated for 15 min. Surplus His$_6$-SUMO-C-peptide was then removed by incubation in G buffer for 15 min. Sensors were then moved to ACD solution and association was measured for 48 min. After that, ACD dissociation was measured in buffer G for 10 min. Measurements were performed with 1000 rpm shaking at 25 °C. Measurements were performed for different ACD concentrations in triplicates. Biolayer thickness at the end of association stage was corrected for the nonspecific binding using control substituting His$_6$-SUMO for His$_6$-SUMO-C-peptide and converted to fraction bound by min-max scaling between 0 and 1 using minimal and maximal triplicate averages of biolayer thickness. To determine dissociation constant fraction bound as a function of ACD concentration was fitted to the Hill equation using SciPy implementation (*Virtanen et al., 2020*) of dogbox algorithm:

$$f_b = \frac{[X]^n}{K_{0.5}^n + [X]^n}$$

Where [X] is the total ACD concentration, $K_{0.5}$-ACD concentration required to reach half-maximum binding at equilibrium, n-Hill coefficient, $f_b$-fraction bound. Standard deviations of fitted $K_{0.5}$ and n values were derived from the diagonal of the optimized parameters covariance matrix.

## Molecular dynamics (MD)

All MD simulations were performed using Gromacs 2019.2 (*Van Der Spoel et al., 2005*) and CHARMM36-jul2021 as a force field (*Huang et al., 2017*). Simulations were performed in the isothermal-isobaric (NPT) ensemble, where the temperature was kept at 310 K using v-rescale thermostat (*Bussi et al., 2007*) with a time constant of 0.1 ps and the pressure was held at 1 bar using Parinello-Rahman barostat (*Parrinello and Rahman, 1981*). Lennard-Jones potential with a cut-off of 1.0 nm was used to describe Van der Waals interactions. Computation of long-range electrostatic interactions was performed using the particle mesh Ewald (PME) method (*Essmann et al., 1995*) with a Fourier grid spacing of 0.12 nm and a real space cutoff of 1.0 nm. Bonds between hydrogen and protein heavy atoms were constrained by P-LINCS (*Hess et al., 1997*), and water molecules geometry was constrained by SETTLE (*Miyamoto and Kollman, 1992*). Integration of equations of motion was performed by leap-frog algorithm (*Van Gunsteren and Berendsen, 1988*) with a time step of 2 fs. Periodic boundary conditions were applied in all dimensions.

The initial conformation of $AncA_0$ ACD-C-terminal peptide complex was predicted by ColabFold implementation (*Mirdita et al., 2022*) of AlphaFold-Multimer (*Evans et al., 2022*) Substitutions Q66H and G109D were introduced to the complex using PyMol Mutagenesis Wizard (http://www.pymol.org). Complexes were then placed in rhombic dodecahedral boxes measuring 10.4 nm in all dimensions and solvated by CHARMM-modified TIP3P water model (*Jorgensen et al., 1983*). Concentrations of sodium and chloride ions were adjusted to 0.15 M and net zero charge of the system. Each system was subjected to a 3-step energy minimization protocol, where during the first step protein conformation was constrained, during the second step constraint was reduced to protein backbone only and during the third step positions of all protein heavy atoms were restrained with a force constant of 1000 kJ*mol$^{-1}$*nm$^{-1}$. Minimized systems were equilibrated for 10 ns while positions of protein backbone atoms were kept constrained, equilibration was then continued without constraints for further 500 ns. The first 100 ns of equilibration was discarded, and the rest was used for ACD-C-terminal peptide contact determination using GetContacts tool (https://getcontacts.github.io/; *Fonseca and Ma, 2018*). Contact between interfacial residues was defined as any of the following types of interaction: hydrogen bond, ionic, π-stacking, π-cation, or van der Waals between purely hydrophobic residues. The default GetContacts interaction criteria were used for all interaction types except for hydrogen bond detection, where a more stringent 30° cutoff for hydrogen-donor-acceptor angle was used. Contact heatmaps were prepared using seaborn (*Waskom, 2021*).

The representative conformations of ACD-C-terminal peptide complexes were chosen by clustering the last 400 ns of equilibrium MD trajectories (frames spaced every 0.5 ns) using 'gmx cluster' tool and Jarvis-Patrick clustering method (*Jarvis and Patrick, 1973*). RMSD cutoff of 0.2 nm, was used for the Jarvis-Patrick algorithm and conformations possessing at least three neighbors in common out of 15 closest conformations were assigned to the same cluster. RMSD calculation was based on coordinates of heavy backbone atoms of the C-terminal peptide interacting ACD monomer without dimerization loop (residues 40–74 and 95–126) and stably interacting region of the C-terminal peptide (residues 132–137). In the case of both simulated complexes the biggest cluster contained more than 90% of simulation frames (92.3% for AncA0 ACD complex and 93.3% for AncA0 Q66H G109D complex) and its middle frame was chosen as representative conformation. Visual Molecular Dynamics (VMD) (*Humphrey et al., 1996*) and Blender (https://www.blender.org/) were used for structure visualization.*Supplementary file 1Supplementary file 3*.

## Acknowledgements

This work was supported by a grant from the Polish National Science Centre (OPUS 17 2019/33/B/NZ1/00352). The work of B.T. and P.D. was supported by a National Science Center grant (OPUS 21 2021/41/B/NZ8/02835)(to B.T.). We gratefully acknowledge Poland's high-performance computing

infrastructure PLGrid (HPC Centers: ACK Cyfronet AGH) for providing computer facilities and support within computational grant no. PLG/2023/016546. We thank Prof. Max Telford, Prof. Jaroslaw Marszalek, and Dr. Agnieszka Kłosowska for their helpful discussions.

## Additional information

### Funding

| Funder | Grant reference number | Author |
|---|---|---|
| Narodowe Centrum Nauki | OPUS 17 2019/33/B/NZ1/00352 | Piotr Karaś<br>Klaudia Kochanowicz<br>Marcin Pitek<br>Igor Obuchowski<br>Krzysztof Liberek |
| Narodowe Centrum Nauki | OPUS 21 2021/41/B/NZ8/02835 | Przemyslaw Domanski<br>Barlomiej Tomiczek |
| Infrastruktura PL-Grid | PLG/2023/016546 | Barlomiej Tomiczek |

The funders had no role in study design, data collection and interpretation, or the decision to submit the work for publication.

### Author contributions

Piotr Karaś, Conceptualization, Data curation, Formal analysis, Investigation, Visualization, Methodology, Writing - original draft, Writing - review and editing; Klaudia Kochanowicz, Investigation, Methodology; Marcin Pitek, Investigation, Visualization, Methodology, Writing - original draft, Writing - review and editing; Przemyslaw Domanski, Investigation, Visualization; Igor Obuchowski, Methodology, Writing - review and editing; Barlomiej Tomiczek, Conceptualization, Formal analysis, Supervision, Validation, Visualization, Writing - original draft, Project administration, Writing - review and editing; Krzysztof Liberek, Conceptualization, Supervision, Funding acquisition, Validation, Writing - original draft, Project administration, Writing - review and editing

### Author ORCIDs

Piotr Karaś ⓘ http://orcid.org/0000-0001-5270-7938
Marcin Pitek ⓘ http://orcid.org/0000-0002-1300-4364
Barlomiej Tomiczek ⓘ http://orcid.org/0000-0001-9295-663X
Krzysztof Liberek ⓘ http://orcid.org/0000-0002-7532-9279

Reviewer #1 (Public Review): https://doi.org/10.7554/eLife.89813.3.sa1
Reviewer #2 (Public Review): https://doi.org/10.7554/eLife.89813.3.sa2
Author Response https://doi.org/10.7554/eLife.89813.3.sa3

## Additional files

### Supplementary files

• Supplementary file 1. Multiple sequence alignment of *Enterobacterales* IbpA orthologs used for phylogenetic analysis and ancestral reconstruction in fasta format.

• Supplementary file 2. Phylogenetic tree of *Enterobacterales* IbpA protein family in newick format (see *Figure 2*).

• Supplementary file 3. Posterior probability statistics for ancestral sequence reconstruction of $AncA_0$ and $AncA_1$ nodes. For each position amino acids reconstructed with posterior probability higher than 0.2 are shown. Single-letter symbols of reconstructed amino acids are followed by posterior probability of reconstruction (in brackets). Positions at which most likely amino acid differ between $AncA_0$ and $AncA_1$ are marked in bold and italics. Posterior probabilities were estimated using the FastML program based on the Maximum Likelihood and Empirical Bayes method.

• Supplementary file 4. Statistics for selection analysis. (A) Branch model statistics for IbpA orthologs

from *Enterobacterales*. Models assumed either branch between nodes AncA$_0$ and AncA$_1$ or entire *Erwiniaceae* clade as foreground; NS – Not significant. (B) Branch-site model statistics for IbpA orthologs from *Enterobacterales*. Models assumed either branch between nodes AncA$_0$ and AncA$_1$ or entire *Erwiniaceae* clade as foreground; NS – Not significant.

• Supplementary file 5. Nucleotide sequences of de novo synthesized genes ordered from Genscript.

• MDAR checklist

## Data availability

All data generated or analysed during this study are included in the manuscript and supporting files. Source data files have been provided for Figs 1, 3, 4, 5, 6 and Figure supplements. Multiple sequence alignment of Enterobacterales IbpA orthologs used for phylogenetic analysis and ancestral reconstruction in fasta format is attached as *Supplementary file 1*. Posterior probability statistics for ancestral sequence reconstruction of AncA$_0$ and AncA$_1$ nodes are attached as *Supplementary file 3*. Raw data generated in this study are attached as source data files.

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
