## [Editor Report · eLife assessment]

This **valuable** study advances our understanding of the evolution of protein complexes and their functions. Through **convincing** experimental and computational methodologies, the authors show that the specialization of protein function following gene duplication can be reversible. The work will be of interest to investigators working in biochemical evolution and those working on heat shock proteins.

---

## [Referee Report · Reviewer #1 (Public Review)]

The work in this paper is in general done carefully. Reconstructions are done appropriately and the effects of statistical uncertainty are quantified properly. I was glad to see that the tree and alignment are now deposited.

The paper identifies which mutations are crucial for the functional differences between the ancestors tested. This is done quite carefully - the authors even show that the same substitutions also work in extant proteins.

These substitutions very slightly lower the affinity and increase the cooperativity of the C-terminal peptide binding to the alpha crystallin domain - a key oligomeric interaction. These relatively minor changes nevertheless apparently affect the subunit exchange behaviour and oligomerization of the sHSP.

Lastly, the authors use likelihood methods to test for signatures of selection. This reviewer is not a fan of these methods, as they are easily misled by common biological processes (see PMID 37395787 for a recent critique). The paper is relatively careful in the interpretation of this test though, and I think the importance of the other findings does not depend on the action of selection along this branch.

---

## [Referee Report · Reviewer #2 (Public Review)]

This was an interesting study, and I enjoyed seeing different experimental approaches used to compare the properties of the different native proteins, the ancestral reconstructions, and the other mutants. In the original manuscript, I felt that the authors had over-simplified their explanations, as the differences between the ancestral proteins, and the changes induced by the two mutations, only partially explain the differences between IbpA proteins from the two different species. However, with their revised version, I think the presentation and discussion of their results are much better. Overall, I think this represents a valuable contribution to the field, providing convincing mechanistic evidence as to how these small heat shock proteins have evolved.

---

## [Author Response]

The following is the authors’ response to the original reviews.

**Reviewer #1 (Recommendations For The Authors):**
My main request is to show the phylogeny in the main text, so the reader knows what nodes are being compared.

Full phylogeny was added to the main text as Fig. 2. Additionally, phylogenetic tree in Newick format is presented as a Supplementary file 2.

I also suggest the authors check their figure legends carefully. At least in figure one, I think there is some mix-up with the letter labelling of the panels.

Our mistake. Figure legend was corrected. In this version of the manuscript Figure 1 was split into Fig. 1 and Fig. 3. Corrected version is presented in the legend to Fig. 3.

And lastly, I urge the authors to deposit the tree, alignment, and reconstructed sequences in a public repository.

Alignment in fasta format and phylogenetic tree in Newick format were added as supplementary files to the publication (supplementary file 1 and supplementary file 2, respectively). Reconstructed sequences(both Most likely and AltAll variants) were shown as a figure supplement (Figure 3 – figure supplement2). Posterior probabilities for all positions of the reconstructed sequences were added as a supplementary file (supplementary file 3).

**Reviewer #2 (Recommendations For The Authors):**
-I find the term "secondarily single sHsp" to be a little confusing, especially because it is often used in relation to IbpA/B, but it is just IbpA in another species. I think it would be more clear for the reader to consistently refer to it as Erwiniaceae IbpA vs Escherichia IbpA, or something similar.

In the introduction we clarified (page 4 lines 11-13) that the term “secondarily single” IbpA refers to IbpA that lacks partner protein as a result of ibpB gene loss. This is in opposition to “single-protein” IbpA from a clade in which gene duplication leading to creation of two – protein sHsp system did not occur (like Vibrionaceae or Aeromonadaceae) - see Obuchowski et al., 2019.

-Figure 1B. The labels are not defined. What is L? A and B refer to IbpA and IbpB but this should be made more clear to the reader. Why is this panel only referred to in the Introduction and not the Results? Why is there a second panel for E.amy, rather than including it in the same panel, as for other experiments? What are the error bars? (That goes for every error bar in the paper, none are defined).

Labels in Fig.1B were corrected; “L” was used in reference to “luciferase alone” and it has been corrected for consistency to “no sHsp”. The sHsps activity measurements (obtained in the same experiment) were split into two separate panels as a correspondence to the two branches of the simplified tree in Fig. 1. The figure was modified to make it clearer and avoid confusion. Definitions of error bars were added to this and other figures.

-"AncA0 exhibited sequestrase activity on the level comparable to IbpA from Escherichia coli(IbpA *E. coli*). AncA1 was moderately efficient in this process and IbpA from Erwinia amylovora(IbpAE.amyl) was the least efficient sequestrase (Fig. 1D)." - First, this should be referring to Fig. 1C. Second, the text doesn't quite match the panel. A0 appears to have the strongest sequestrase activity over most concentrations. Can the authors comment on in what concentration range these differences are most meaningful?

Figure legend was corrected. Descriptions of panels C and D were fixed. Now these data are presented in panels A and B of a new Fig. 3. In our opinion differences in sequestration are most meaningful at lower sHsp concentrations (in this case lower than 5 µM), as with high enough sHsp concentration even less effective sequestrases seem to be able to effectively sequester aggregated proteins. Comment about it was added to the main text (page 5, line 6)

-"Ancestral proteins' interaction with the aggregated substrates was stronger than in the case of extant E. amylovora IbpA, but weaker than in the case of extant *E. coli* IbpA (Fig. 1C)." - Is this referring to Fig.1C, or to the unlabelled panel on the bottom right panel of Fig 1 (that is not referred to in the legend)? Can the authors comment on why they think the 2 ancestral proteins are much more similar to each other than they are to either of the native IbpAs?

Due to our mistake descriptions of panels C and D were switched.

Figure 1 was rearranged and split into Figures 1 and 3. Former figure S1 (full phylogeny) was inserted into the main text, as Fig. 2, per request of reviewer #1. Former panel 1D (now 3B) was rearranged, as graph was not apparent to be a part of that panel and looked as if it was unlabeled.

The fact that the two ancestral proteins are more similar to each other than to the extant *E. coli* and E. amylovora proteins in their interaction with model substrate might be caused by higher sequence identity between the two ancestral proteins than between ancestral and extant proteins (10 amino acid differences between AncA0 and AncA1 compared to 20 differences between AncA1 and IbpA from E. amylovora or 11 differences between AncA0 and IbpA from E. coli). One also has to remember that this property is only one aspect of sHsp activity – proteins AncA0 and AncA1 are much less similar to each other if other activities such as sequestrase activity are considered. Substrate affinity and sequestrase activity are connected to each other, but there isn’t a strict correlation, as can be seen in the case of free ACD domains, which strongly bind aggregated substrate while effectively lacking sequestrase activity (fig. 5 A, fig. 5 – figure supplement 4 A,B).

-Figure 1E should have E. coli IbpA and IbpB, by themselves, included for comparison. Strangely, it seems, by comparison to Fig 1B, that the "inhibitory" activity of A0 is not present in the *E. coli* protein, and the authors should comment on this. Similarly, A1 disaggregation looks like it might not be significantly different than the E. coli protein. Can the authors comment on why disaggregation might be so low in A1 compared to E.amy?

*E. coli* IbpA alone was added to Fig. 1E (Fig. 3C in the new version) as suggested.

AncA1 indeed exhibits similar activity to extant IbpA from *E. coli*, which, at the conditions of the experiment, does not possess inhibitory effect observed for AncA0. This suggests that:

-There was an additional increase in ability to stimulate luciferase disaggregation between AncA1 and extant IbpA from E. amylovora

-There was also an increase of ability to stimulate luciferase refolding between AncA0 and extant *E. coli* IbpA, albeit to a significantly lesser degree than in the Erwiniaceae branch.

It is quite likely that after separation of Erwiniaceae and Enterobacteriaceae sHsp systems, they underwent further optimization through evolution. This might have led to observed higher effectiveness of modern IbpAs from both clades in refolding stimulation in comparison to the reconstructed ancestral proteins.

Despite the above, effects of substitutions on positions 66 and 109 on activities of the extant *E. coli* and E. amylovora proteins suggests that the two identified positions still play key role in differentiating extant IbpAs from Erwiniaceae and Enterobacteriaceae.

Nevertheless, additional mutations that lead to increased ability to stimulate luciferase reactivation must have occurred in both Erwiniaceae and Enterobacteriaceae branches of the phylogeny during evolution. These substitutions would be a worthwhile subject of further study.

-Fig 1D - lizate should be lysate.

The typo was corrected.

-What is the bottom right panel in Fig 1? It doesn't seem to be referred to in the legend.

This panel was intendent to be the part of figure 1D, but it was not clearly visible. This figure was rearranged to make it clearer. Now these data are presented as Fig. 3B.

-Sequences are provided for the ancestral proteins, but I don't see them anywhere for the alternative ancestral proteins. How similar are the Anc proteins to the AltAlls? If they are very similar, this may not tell us anything about "robustness".

Sequences of alternative proteins are added as a figure supplement (Fig. 3 - figure supplement 2). Full sequences of ML and alternative ancestors with posterior probabilities for each reconstructed position are presented in supplementary file 3

The testing of the robustness to statistical uncertainty was intended to test to what extent properties of reconstructed ancestral proteins could be influenced by uncertainty present in a given reconstruction due to probabilistic nature of the process. Relatively high similarity between ML and AltAll sequences would indicate low uncertainty of the reconstruction (most likely due to high conservation during evolution). In such a case similar properties of AltAll and ML proteins would simply indicate that they are robust to the level of uncertainty present in a given reconstruction (which may be low). It would not tell us much about “general” robustness to mutations, but it was not relevant to research questions considered.

-If the functional gain by IbpA comes down to only two amino acid substitutions, I'm not convinced this would be meaningfully reflected in any tests of positive selection.

After considering Reviewer #1’s comments about limitations of models used for selection analysis we added acknowledgment in the discussion (page 9, line 9 - 13) that results indicating positive selection in our dataset should not be considered conclusive (see answer to Reviewer #1’s public review below).

-The full MSA should be provided as supplemental material.

The full MSA in fasta format is presented in the supplementary file 1.

-For the aggregate binding panels in Figs 3 and 4, it would be helpful to show the native and ancestral proteins for comparison. I know this is a bit redundant, as they're present in Fig 1, but I find it hard to judge the scale of change. This is especially important because A0 and A1 are very similar in Fig 1, so I want to see what kind of difference the 2 mutations make.

Data presented in Fig. 3C (Fig. 5C in the new version) refer to the binding of α-crystallin domains (A0ACD and A0ACD Q66H G109D) and not full length sHsps to *E. coli* proteins aggregated on a BLI sensor. Our intention was to show the influence of the two crucial substitutions (Q66H G109D) on the properties of A0 ancestral α-crystallin domain.

Figure 4 (Fig. 6 in the new version) represent the effects of the substitutions on the identified positions 66 and 109 on the properties of extant IbpA orthologs from *E. coli* and E. amylovora, showing that these two positions play a key role in differentiating properties of those extant proteins. Changes in binding to aggregated substrate caused by those substitutions, as shown in Figure 6 B,C (new version), are indeed larger than observed between AncA0 and AncA1, as shown in Fig. 3B (new version).

One has to remember, however, that the experiment shown in Fig.3 (new version) shows the effects of all 10 amino acid changes between the nodes A0 and A1 and not only the two analyzed substitutions, as was the case in experiment shown in Fig. 6 B,C (new version). Moreover, due to relatively large number of differences between ancestral and extant sequences (11 differences between AncA0 and *E. coli* IbpA, 20 differences between AncA1 and E. amylovora IbpA), substitutions in the two experiments are introduced into different sequence context.

Because of the above, we believe that direct comparison of the results obtained for ancestral proteins with the results obtained for substitutions introduced into extant proteins would not meaningfully contribute to answering the question of the role of analyzed substitution in the context of extant proteins, while decreasing clarity of presented information.

-Some of the luciferase plots show a time course, but others just show a single %. What is the time point used for the single % plots?

Information was added to appropriate figure legends that for experiments showing a single timepoint the luciferase activity was measured after 1h of refolding.

**Reviewer #3 (Recommendations For The Authors):**
1. In the Introduction, it would be beneficial to explore additional instances where this evolutionary simplification process has been observed in nature. Investigating the prevalence of this phenomenon and identifying other multi-protein systems that have undergone simplification could enhance the understanding of its significance and implications.

The section of the introduction concerning gene loss and differential paralog retention was expanded with additional examples of gene loss that is considered adaptive (page 3 lines 1 - 12).

1. I am intrigued by the reasons why certain organisms continue to maintain a two-protein system despite the viability of a single-protein system. This aspect is particularly relevant for bacteria, considering the fitness cost associated with maintaining extra gene copies. Do you have any hypotheses or theories that may shed light on this intriguing observation?

Refolding of proteins from aggregates requires the functional cooperation of sHsps and chaperones from Hsp70 system and Hsp100 disaggregase. In two protein sHsps system one sHsp (IbpA) is specialized in substrate binding, while the second one (IbpB) possesses low substrate binding potential and enhances sHps dissociation from substrates (Obuchowski et al, 2019). Thus, the presence of IbpB reduces the amount of chaperones from Hsp70 system required to outcompete sHsps from aggregated substrates to initiate refolding process. The cost associated with maintaining extra sHsp gene copy (ibpB) in bacteria might be compensated by lower requirement for Hsp70 chaperones for efficient and fast protein refolding following stress conditions.

In this study we have demonstrated how such a system could have been simplified to a single – protein system capable of efficient substrate sequestration as well as stimulation of reactivation. This indeed leads to the question why such single – protein system isn’t more prevalent in Enterobacterales.

One possibility may be that there are very specific requirements for efficient reactivation by a single – protein sHsp system. We have shown that new, more efficient IbpA functionality observed in Erwiniaceae required at least two separate mutations. It is possible, that such combinations of two substitutions simply did not occur in Enterobacteriaceae clade, in which IbpA still required partner protein for efficient reactivation stimulation.

One must also remember that experiments performed in this study were performed in vitro in a specific set of conditions, which most likely does not represent whole spectrum of challenges faced by different bacteria. It is possible that two – protein system has some other additional adaptive effects, counterbalancing the additional cost of gene maintenance. It was for example recently shown (Miwa & Taguchi, PNAS, 120 (32) e2304841120) that bacterial sHsps play an important role in regulation of stress response. Two – protein system could potentially allow for more complex regulation.

1. Incorporating X-ray crystallization as an additional technique in the methodology would offer detailed molecular insights into the effects of Q66H and G109D substitutions on ACD-C-terminal peptide and ACD-substrate interactions. The inclusion of such data would further strengthen the results section and provide robust support for your findings. Since the x-ray data might be difficult to collect, the authors might think to get alphafold model or some rosetta score for the model to discuss the finding further.

In response to reviewer comment we added the comparison of the structural models of AncA0 and AncA0 Q66H G109D ACD dimers complexed with the C-terminal peptides, representing middle structures of largest clusters obtained from equilibrium molecular dynamics simulation trajectories based on the AlphaFold2 prediction and in silico mutagenesis (Fig. 5 – figure supplement 2). Model comparison as well as C-terminal peptide – ACD contact analysis did not reveal any major changes in mode of peptide binding or α-crystallin domain conformation, although we do acknowledge that simulation timescale limits the conformational sampling.

**Reviewer #1 (Public Review):**
The work in this paper is in general done carefully. Reconstructions are done appropriately and the effects of statistical uncertainty are quantified properly. My only slight complaint is that I couldn't find statistics about posterior probabilities anywhere and that the sequences and trees do not seem to be deposited.

Posterior probabilities for all positions of reconstructed proteins were added as a supplementary file 3. MSA of all sequences used for ancestral reconstruction as well as phylogenetic tree in Newick format were added as supplementary files 1 and 2, respectively.

I would also have preferred to have the actual phylogeny in the main text. This is a crucial piece of data that the reader needs to see to understand what exactly is being reconstructed.

Full phylogeny was added to the main text as Fig. 2.

The paper identifies which mutations are crucial for the functional differences between the ancestors tested. This is done quite carefully - the authors even show that the same substitutions also work in extant proteins. My only slight concern was the authors' explanation of what these substitutions do. They show that these substitutions lower the affinity of the C-terminal peptide to the alpha-crystallin domain - a key oligomeric interaction. But the difference is very small - from 4.5 to 7 uM. That seems so small that I find it a bit implausible that this effect alone explains the differences in hydrodynamic radius shown in Figure S8. From my visual inspection, it seems that there is also a noticeable change in the cooperativity of the binding interaction. The binding model the authors use is a fairly simple logarithmic curve that doesn't appear to consider the number of binding sites or potential cooperativity. I think this would have been nice to see here.

The binding model we used is equivalent to the Hill equation as it accounts for the variable slope of sigmoid function by inclusion of input scaling factor k, which is equivalent to the hill coefficient. Simple one site binding model and two site binding model were also considered but provided worse fits to the data than model including binding cooperativity. Not providing values of fitted parameter k was our mistake, and it was corrected (Fig. 5. with a legend). Additionally, output scaling parameter L is not necessary as fraction bound takes values from 0 to 1, therefore we have fitted the curves again without this parameter. The new values of fitted parameters are very similar to the previous ones. To make text more accessible to the reader, we have used a conventional form of Hill equation. Indeed, AncA0 Q66H G109D ACD displays higher binding cooperativity than more ancestral AncA0 ACD (hill coefficient 2.3 for AncA0 vs 3.7 for AncA0 Q66H G109D). Fitted values of Hill coefficients are higher than one can expect for 2-site ACD dimer, which is probably caused by an experimental setup of BLI, where C-terminal peptide is immobilized on the sensor and ACD is present in solution as bivalent analyte leading to emergence of avidity effects. Both cooperativity and avidity are reflected in the value of Hill coefficient, however as ligand density on the sensor is the same in all experiments only change in ACD binding cooperativity can account for observed difference in the value of Hill coefficients. Difference in the C-terminal peptide binding cooperativity may influence the process of sHsp oligomerization and assembly formation despite similar binding affinity, especially if avidity of multiple binding sites within oligomer is considered.

In addition, we changed the legend to Figure S8 (now called Fig. 5 – figure supplement 4A ) to clarify the fact that the differences in average hydrodynamic radius are in fact ferly small. To highlight the observation that there are two populations of particles in AncA0 and AncA0 Q66H G109D measured at 25, 35 and 45 °C with different hydrodynamic diameters, we used % of intensity in DLS measurement. It allows us to show the change in the hydrodynamic diameter distribution that is relatively small. We recognize it was not properly explained in the article and added a clarification in figure description.

Lastly, the authors use likelihood methods to test for signatures of selection. This reviewer is not a fan of these methods, as they are easily misled by common biological processes (see PMID 37395787 for a recent critique). Perhaps these pitfalls could simply be acknowledged, as I don't think the selection analysis is very important to the impact of the work.

We thank the reviewer for pointing to the recent research about limitations of methods used in our work in selection analysis. As per recommendation we added acknowledgment of limitations of methods used to discussion (page 9, line 9 - 13), modifying wording of our conclusions to deemphasize significance of selection analysis results.